# DEEP GENERATIVE DUAL MEMORY NETWORK FOR CONTINUAL LEARNING

## ABSTRACT

Despite advances in deep learning, artificial neural networks do not learn the same way as humans do. Today, neural networks can learn multiple tasks when trained on them jointly, but cannot maintain performance on learnt tasks when tasks are presented one at a time – this phenomenon called catastrophic forgetting is a fundamental challenge to overcome before neural networks can learn continually from incoming data. In this work, we derive inspiration from human memory to develop an architecture capable of learning continuously from sequentially incoming tasks, while averting catastrophic forgetting. Specifically, our model consists of a dual memory architecture to emulate the complementary learning systems (hippocampus and the neocortex) in the human brain and maintains a consolidated long-term memory via generative replay of past experiences. We (i) substantiate our claim that replay should be generative, (ii) show the benefits of generative replay and dual memory via experiments, and (iii) demonstrate improved performance retention even for small models with low capacity. Our architecture displays many important characteristics of the human memory and provides insights on the connection between sleep and learning in humans.

## 1 INTRODUCTION

Many machine learning models, when trained sequentially on tasks, forget how to perform the previously learnt tasks. This phenomenon called *catastrophic forgetting* is prominent in neural networks (McCloskey & Cohen, 1989). Without a way to avert catastrophic forgetting, a learning system needs to store all training data and relearn on it along with new incoming data, when retraining. Hence, it is an important challenge to overcome in order to enable systems to learn continuously.

McCloskey & Cohen (1989) first suggested that the underlying cause of forgetting was the distributed shared representation of tasks via network weights. Subsequent works attempted to remedy the issue by reducing representational overlap between input representations via activation sharpening algorithms (Kortge, 1990), orthogonal recoding of inputs (Lewandowsky, 1991) or orthogonal activations at all hidden layers (McRae & Hetherington, 1993; French, 1994). More recent works have explored activations like dropout (Goodfellow et al., 2015) and local winner-takes-all (Srivastava et al., 2013) to create sparse, less correlated feature representations. But such sparse encodings can be task specific at times and in general act as heuristics to mildly pacify the underlying problem.

Further, natural cognitive systems are also connectionist in nature and yet they forget gradually but not 'catastrophically'. For instance, humans demonstrate gradual systematic forgetting. Frequently and recently encountered tasks tend to survive much longer in the human memory, while those rarely encountered are slowly forgotten. Some of the earlier tasks may be seen again, but it is not necessary for them to be retained in memory (French, 1999). Hence only sparsifying representations does not solve the problem. Instead, neuroscientific evidence suggests that humans have evolved mechanisms to separately learn new incoming tasks and consolidate the learning with previous knowledge to avert catastrophic forgetting (McClelland et al., 1995; O'Neill et al., 2010; French, 1999).

**Complementary learning systems**: McClelland et al. (1995) suggested that this separation has been achieved in the human brain via evolution of two separate areas of the brain, the hippocampus and the neocortex. The neocortex is a long term memory which specializes in consolidating new information with previous knowledge and gradually learns the joint structure of all tasks and experiences; whereas

the hippocampus acts as a temporary memory to rapidly learn new tasks and then slowly transfer the knowledge to neocortex after acquisition.

**Experience replay**: Another factor deemed essential for sequential learning is experience replay. McClelland et al. (1995); O'Neill et al. (2010) have emphasized the importance of replayed data patterns in the human brain during sleep and waking rest. Robins (1995; 2004) proposed several replay techniques (a.k.a. pseudopattern rehearsal) to achieve replay, but they involved generating replay data without storing input representations and our experiments show that they lack the accuracy required for consolidation.

**Weight consolidation or freezing**: Recent evidence from neuroscience also suggests that mammalian brain protects knowledge in the neocortex via task-specific consolidation of neural synapses over long periods of time (Yang et al., 2014; Benna & Fusi, 2016). Such techniques have recently been employed in progressive neural networks (Rusu et al., 2016) and Pathnets (Fernando et al., 2017) both of which freeze neural network weights after learning tasks. Kirkpatrick et al. (2017) have used the fisher information matrix (FIM) to slow down learning on network weights which correlate with previously acquired knowledge.

In this paper, we address the catastrophic forgetting problem by drawing inspiration from the above neuroscientific insights and present a method to overcome catastrophic forgetting. More specifically, we propose a dual-memory architecture for learning tasks sequentially while averting catastrophic forgetting. Our model comprises of two generative models: a short-term memory (STM) to emulate the human hippocampal system and a long term memory (LTM) to emulate the neocortical learning system. The STM learns new tasks without interfering with previously learnt tasks in the LTM. The LTM stores all previously learnt tasks and aids the STM in learning tasks similar to previous tasks. During sleep/down-time, the STM generates and transfers samples of learnt tasks to the LTM. These are gradually consolidated with the LTM's knowledge base of previous tasks via generative replay.

Our approach is inspired from the strengths of deep generative models, experience replay and the complementary learning systems literature. We demonstrate our method's effectiveness in averting catastrophic forgetting by sequentially learning multiple tasks. Moreover, our experiments shed light on some characteristics of human memory as observed in the psychology and neuroscience literature.

## 2 PROBLEM SETTING: SEQUENTIAL MULTITASK LEARNING

Formally, our problem setting can be called *Sequential Multitask Learning* and is characterized by a set of tasks $\mathbb{T}$, which are to be learnt by a model parameterized by weights $\theta$ (e.g. a neural network). From here on, we will use the the phrase *model* and *neural network* interchangeably. In this work we mainly consider supervised learning tasks i.e. task $t \in \mathbb{T}$ has training examples: $\{x_i^t, y_i^t\}_{i=1:N_t}$ for $x_i^t \in \mathcal{X}$ and $y_i^t \in \mathcal{Y}$, but our model easily generalizes to unsupervised learning settings. Note that tasks are presented sequentially and the total number of tasks $|\mathbb{T}|$ is not known a priori.

**Finite memory**: We further assume that any training algorithm can store some examples from each task if needed, but the storage ($N_{max}$) is limited and can be smaller than the total number of examples from all tasks $\left(\sum_{t=1}^{|\mathbb{T}|} N_t\right)$. So, algorithms cannot store all training examples and re-learn on them when new tasks arrive. The same restriction applies to algorithms with generative models i.e. no more than $N_{max}$ examples allowed at any time (generated + stored).

For testing, the model can be asked to predict the label $y^t \in \mathcal{Y}$ for any example $x^t \in \mathcal{X}$ from any previously seen task $t \in \mathbb{T}$. Our goal is to devise an algorithm which learns these tasks sequentially while avoiding catastrophic forgetting and can achieve a test loss close to that of a model which learnt all the tasks jointly.

## 3 DEEP GENERATIVE DUAL MEMORY NETWORK

The idea of replaying experience to a neural network has been used previously for reinforcement learning (Lin, 1993; Mnih et al., 2015). A study by O'Neill et al. (2010) suggests that experience replay also occurs in the human brain during sleep and waking rest and aids in consolidation of learnt experiences. We propose that experience replay must be generative in nature. This is better than storing all samples in replay memories as is common in reinforcement learning (Mnih et al., 2015),

since sampling from a generative model automatically provides the most frequently encountered samples. It is also feasible with limited total memory, whereas explicitly storing samples from previous tasks requires determining which and how many samples to store for each task. Determining this can depend on the total tasks $|\mathbb{T}|$, number of examples per task $N_t$ and frequency of occurrence of samples, which are often not available a priori.

Previously proposed non-generative approaches to experience replay (Robins, 1995; French, 1997; Robins, 2004) propose to preserve neural networks' learnt mappings by arbitrarily sampling random inputs and their corresponding outputs from the neural networks and using them along with new task samples while training. These approaches have only been tested in small binary input spaces in previous works, and our experiments in section 4 show that sampling random inputs in high-dimensional spaces (e.g. images) does not preserve the mapping learnt by neural networks.

### 3.1 GENERATIVE EXPERIENCE REPLAY

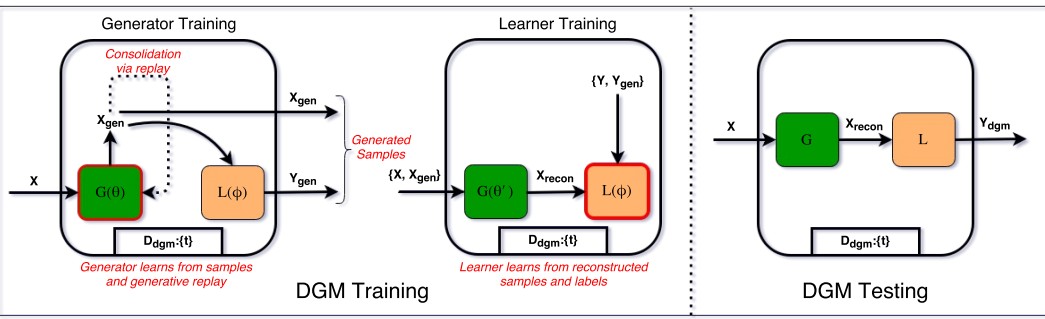

Figure 1: Deep Generative Replay to train a Deep Generative Memory

**Deep Generative Memory (DGM)**: We first introduce a sub-model called the Deep Generative Memory (see figure 1) which has three elements: (i) a generative model (the generator $G$), (ii) a feedforward network (the learner $L$), and (iii) a dictionary ($D_{dgm}$) with task IDs of learnt tasks and the number of times they were encountered. We call this a memory because of its weights and learning capacity, not due to any recurrent connections. We assume availability of unique task IDs for replay and to identify repetition. In practice, a task identification system (e.g., a HMM-based inference model) like in previous works (Kirkpatrick et al., 2017) suffices for this purpose. We choose variational autoencoder (VAE) (Kingma & Welling, 2014) for the generator, since our generative model requires reconstruction capabilities (see section 3.2).

**Deep Generative Replay (DGR)**: We update a DGM with samples from (multiple) new tasks using our algorithm Deep Generative Replay (see figure 1 above and algorithm 1 in appendix A). Given new incoming samples $(X, Y)$, DGR first computes the fraction of total samples that should come from incoming samples ($\eta_{tasks}$) and the fraction to come from previous task samples ($\eta_{gen}$) proportionate to the number of tasks (counting repetitions). It allots a minimum fraction $\kappa$ of the memory capacity $N_{max}$ per new task. This ensures that as the DGM saturates with tasks over time, new tasks are still learnt at the cost of gradually losing performance on the least recent previous tasks. This saturation is synonymous to how learning slows down in humans as they age but they still continue to learn new tasks while forgetting old things gradually (French, 1999). Next, DGR computes the number of samples to be generated from previous tasks and subsamples the incoming samples (if needed) to obey maximum memory capacity ($N_{max}$). It then generates samples of previously learnt tasks $(X_{gen}, Y_{gen})$ using the generator and learner, reconstructs the data $\{X, X_{gen}\}$ using the generator (hence we use a VAE) and then trains the DGM on resulting samples $(X_{recon}, \{Y, Y_{gen}\})$. Doing this final reconstruction provides robustness to noise and occlusion (section 5).

### 3.2 DUAL MEMORY NETWORKS

A good continual learning system needs to quickly acquire new tasks and also retain performance on previously learnt tasks. These conflicting requirements are hard to satisfy simultaneously. Hence, inspired by nature's solution to this problem, we propose a dual memory network to combat forgetting.

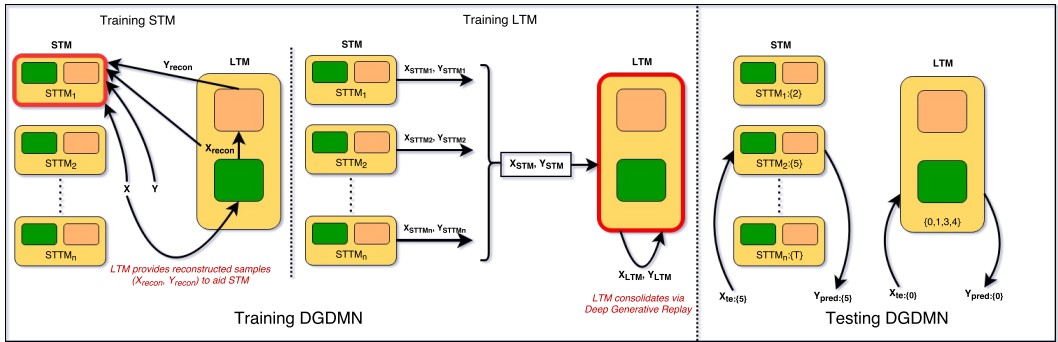

Figure 2: Deep Generative Dual Memory Network (DGDMN)

Our architecture (DGDMN) shown in figure 2 comprises of a large deep generative memory (DGM) called the long-term memory (LTM) which stores information of all previously learnt tasks like the neocortex, and a short-term memory (STM) which behaves similar to the hippocampus and learns new incoming tasks quickly without interference from previous tasks. The STM is a collection of small, dedicated, task-specific deep generative memories (called short-term task memory – STTM), which can each learn one unique task. If an incoming task comes is already in an STTM, the same STTM is used to retrain on it, otherwise a fresh STTM is allocated to the task. Additionally, if the task has been previously consolidated then the LTM reconstructs the incoming samples for that task using the generator (hence we use a VAE), predicts labels for the reconstructions using its learner and sends these newly generated samples to the STTM allocated to this task. This provides extra samples on tasks which have been learnt previously and helps to learn them better, while also preserving the previous performance on that task to some extent.

Once all $(n_{STM})$ STTMs are exhausted, the architecture sleeps (like humans) to consolidate all tasks into the LTM and free up the STTMs for new tasks. While asleep, the STM generates and sends samples of learnt tasks to the LTM, where these are consolidated via deep generative replay (see figure 2). While testing on task $t$ (even intermittently between tasks), if any STTM currently contains task $t$, it is used to predict the labels, else the prediction is deferred to the LTM. This allows predicting on all tasks seen uptil now (including the most recent ones) without sleeping.

## 4 EXPERIMENTS

We perform experiments to demonstrate forgetting on sequential image classification tasks. We briefly describe our datasets here (details in appendix B): (a) **Permnist** is a catastrophic forgetting (Goodfellow et al., 2015; Kirkpatrick et al., 2017) benchmark and each task contains a fixed permutation of pixels on MNIST images (LeCun et al., 1998), (b) **Digits** dataset involves classifying a single MNIST digit per task, (c) **TDigits** is a transformed variant of MNIST similar to Digits but with 40 tasks for long task sequences, (d) **Shapes** contains several geometric shape classification tasks, and (e) **Hindi** contains a sequence of 8 tasks with hindi language consonant recognition.

Along with our model (DGDMN), we test several baselines for catastrophic forgetting, which are briefly described here (implementation and hyperparameter details in appendix B): (a) **Feedforward neural networks (NN)**: We use these to characterize the forgetting in the absence of any prevention mechanism and as a datum for other approaches, (b) **Neural nets with dropout (DropNN)**: Goodfellow et al. (2015) suggested using dropout as a means to prevent representational overlaps and pacify catastrophic forgetting, (c) **Pseudopattern Rehearsal (PPR)**: A non-generative approach to experience replay (Robins, 2004), (d) **Elastic Weight Consolidation (EWC)**: Kirkpatrick et al. (2017) proposed using the Fisher Information Matrix for task-specific consolidation of weights in a neural network, and (e) **Deep Generative Replay (DGR)**: Using a single DGM to separate the effects of generative replay and dual memory architecture.

In our preliminary experiments, we observed that large networks with excessive parameters can more easily adapt to sequentially incoming tasks, thereby masking the severity of catastrophic forgetting. So we have chosen network architectures which have to share all their parameters appropriately amongst

the various tasks in a dataset to achieve reasonable *joint* accuracy on the dataset. This allows us to evaluate an algorithm carefully while ignoring the benefits provided by excessive parameterization.

## 4.1 ACCURACY AND FORGETTING CURVES

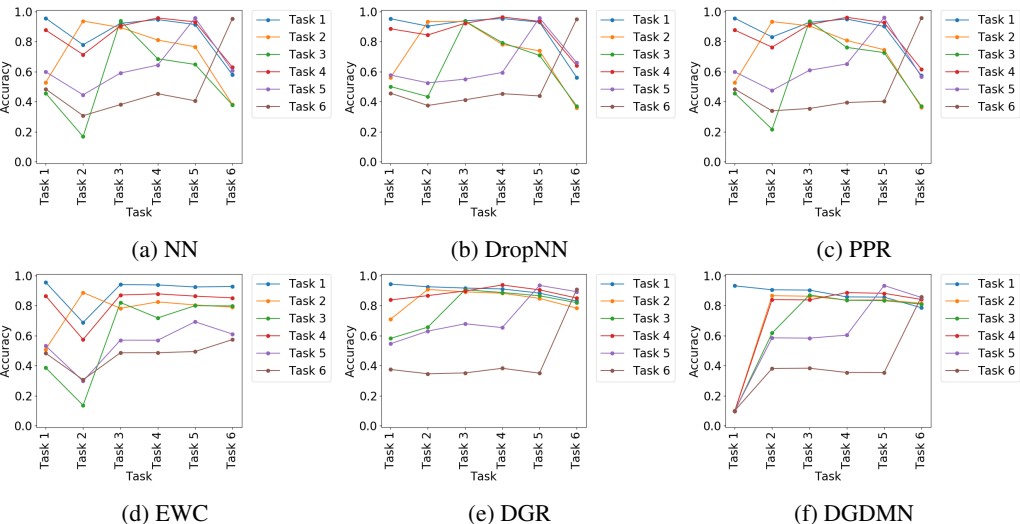

Figure 3: Accuracy curves for Permnist (x: tasks seen, y: classification accuracy on task).

We trained DGDMN and all above baselines sequentially on the image classification tasks of Permnist, Digits, Shapes and Hindi datasets (separately). We show results on the Shapes and Hindi dataset in appendix A. The classification accuracy on a held out test set for each task, after training on the $t^{th}$ task has been shown in figures 3 and 4. We used the same network architecture for each of NN, PPR, EWC, learner in DGR, and learner in the LTM of DGDMN (for a single dataset). DropNN had two intermediate dropout layers after each hidden layer (see appendix B for details).

We observe from figures 3a and 3b, that NN and DropNN forget catastrophically when they learn new tasks. This shows that sparse representation based methods rely on the neural network being of high enough capacity to learn sparse representations (Goodfellow et al., 2015) and may not perform well if the network does not have redundant weights available. EWC forgets less than NN and DropNN, but it rapidly slows down learning on many weights and its learning effectively stagnates after Task 3 (e.g. see Tasks 5 and 6 in figure 3d). The learning slowdown on weights hinders EWC from reusing those weights later on to jointly discover common structures amongst previously learnt and newly incoming tasks. Note that the networks do have the capacity to learn all tasks and our algorithms DGR and DGDMN outperform all baselines by learning all tasks sequentially with this same learner network (figures 3e, 3f).

We observed heavy forgetting on Digits (figure 4) for most baselines, which is expected because all samples in the $t^{th}$ task have a single label ($t$) and so the $t^{th}$ task can be learnt on its own by setting the $t^{th}$ bias of the softmax layer to be high and the other biases low. Such sequential tasks cause catastrophic forgetting. We observed that NN, DropNN, PPR and EWC learnt only the task being trained on and forgot all previous knowledge immediately. Sometimes, we also observed saturation due to the softmax bias being set very high and then being unable to recover from it. PPR showed severe saturation since its replay prevented it from coming out of the saturation.

DGR and DGDMN still retain performance on all tasks of Digits, and our replay strategy prevents saturation by appropriately balancing the ratios of new incoming samples and generated samples from previous tasks. The average forgetting on all tasks $\in \{1, \ldots, t\}$, after training on the $t^{th}$ task (for both Digits and Permnist) is shown in figure 5. For absolute reference, the accuracy of NN by training it jointly on all tasks uptil the $t^{th}$ task has also been shown for each $t$. Again DGR and DGDMN outperform baselines in terms of retained average accuracy. In figure 5b, NN, DropNN, PPR and EWC follow nearly overlapping curves ($acc \approx \frac{1}{t}$) since they are only able to learn one task at a time. Further, though PPR involves experience replay, it does not compare against DGR and

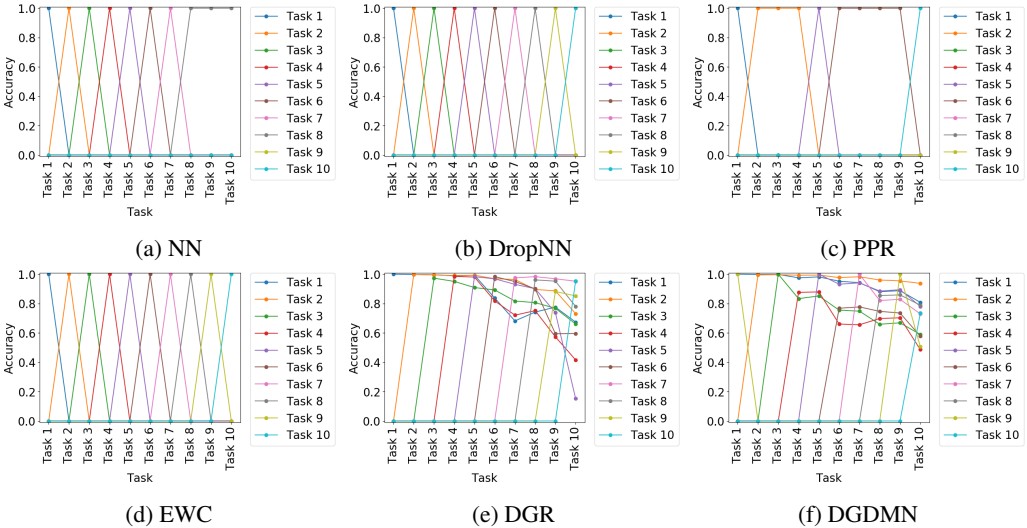

Figure 4: Accuracy curves for Digits (x: tasks seen, y: classification accuracy on task).

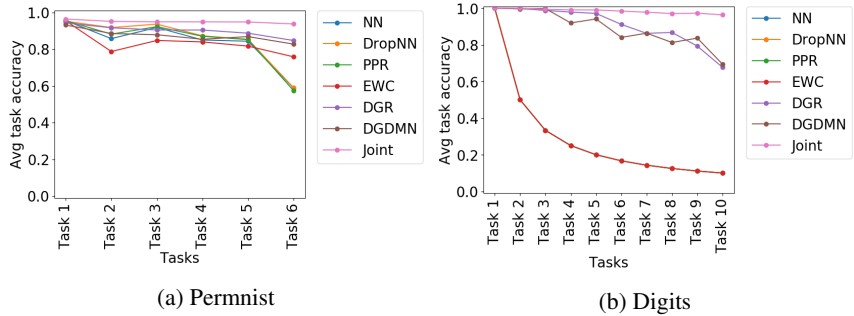

Figure 5: Forgetting curves (x: tasks seen, y: avg classification accuracy on tasks seen).

DGDMN (figures 3c, 4c). Although, it does preserve its learnt mapping around the points randomly sampled from its domain, these random samples are not close to real images and fail to preserve performance. These observations substantiate our claim that any replay mechanism must model the input domain accurately and hence needs to be generative in nature. We observed similar results for the Shapes and Hindi dataset (appendix A).

We point out that datasets like Digits, which contain tasks with highly correlated input (and/or output) samples should be important benchmarks for continual learning for two main reasons: (i) High correlation amongst task samples promotes overfitting to the new incoming task and therefore causes catastrophic forgetting. Being able to retain performance on such task sequences is a strong indicator of the efficacy of a continual learning algorithm. (ii) Humans also learn by seeing many correlated samples together in a short span of time, rather than witnessing nearly IID samples (like in Permnist). For examples, kids learn a single alphabet per day in kindergarten by seeing and writing that alphabet many times that day. Since NN, DropNN and PPR do not fare well on such tasks, we show experiments on EWC, DGR and DGDMN from here on.

## 4.2 REPEATED TASKS AND REVISION

It is well known in psychology literature that human learning improves via revision (Kahana & Howard, 2005; Cepeda et al., 2006). We show performance of EWC and DGDMN on Permnist, when some tasks are repeated (figure 6). DGR performs very similar to DGDMN, hence we omit it. EWC stagnates and once learning has slowed down on the weights important for Task 1, the weights cannot be changed again, not even for improving Task 1. Further, it did not learn Task 6 the first time and

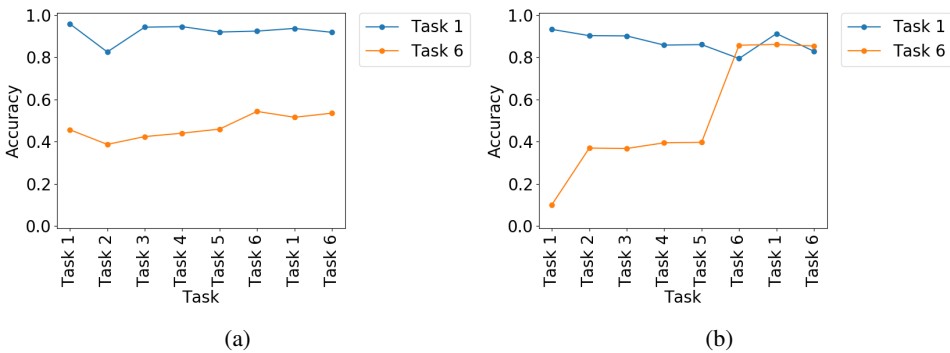

Figure 6: Accuracy curves when tasks are revised: (a) EWC, (b) DGDMN.

revision does not help either. However, DGDMN learns all tasks uptil Task 6, then benefits by revising Task 1 again (accuracy goes up), and somewhat for Task 6 (it did not forget Task 6 substantially). We reiterate that DGDMN, by its design, benefits significantly from revision because STTMs learning a repeated task gain extra samples from the LTM (or generated samples from themselves, if they had learnt the task before). While many previous works do not investigate revision, it is crucial for learning continuously and should improve performance on tasks. The ability to learn from correlated task samples and revision makes our memory architecture functionally similar to that of humans.

### 4.3 CONNECTIONS TO COMPLEMENTARY LEARNING SYSTEMS AND SLEEP

To explore the role of the dual memory architecture and differentiate between DGDMN and DGR, we trained these algorithms on the long sequence of $40$ tasks from TDigits dataset. We limited $N_{max}$ to $120,000$ samples for this task to explore the case where the LTM in DGDMN (DGM in DGR) cannot regenerate as many samples as in the full dataset and has to forget some tasks. At least $\kappa = 0.05$ fraction of memory was ensured per new task and consolidation in DGDMN happened after $n_{STM} = 5$ tasks.

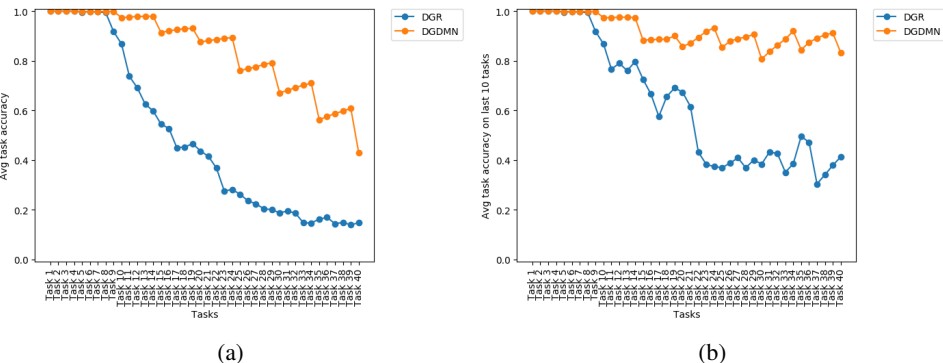

Figure 7: Accuracy curves for TDigits on: (a) tasks seen so far, (b) last 10 tasks seen.

The average forgetting curves vs. tasks encountered are plotted in figure 7a. DGDMN and DGR start around an average accuracy of $1.0$, but start dropping after $10$ tasks since the LTM (DGM for DGR) begins to saturate. While DGDMN drops slowly and retains $> 40\%$ accuracy on all tasks, DGR drops below $20\%$ accuracy. This is because DGR consolidates its DGM too often and the DGM's self-generated slightly erroneous samples compound errors quite fast. DGDMN uses small STTMs to learn single tasks with low error and transfers them simultaneously to the LTM. As a consequence, DGDMN consolidates its LTM with more accurate samples and less often, hence its error accumulates much slower. We discuss the effect of the small error in STTM representations in section 5.

Even though DGDMN displays inevitable forgetting in figure 7a (due to memory constraint), the forgetting is gradual and not catastrophic as seen for NN, DropNN, PPR etc. on Digits dataset. We also measure average accuracy on the most recent few tasks seen (say 10). Figure 7b shows that DGDMN oscillates around $90\%$ average accuracy, whereas DGR's frequent consolidation propagates errors too fast and its accuracy drops even on this metric.

Another advantage of dual memories is revealed by the training time for the algorithms. Figure 9a shows an order of magnitude of difference between DGDMN and DGR in training time. This is because STTMs are smaller and faster to train than the LTM. LTM preserves all the tasks seen so far and hence requires a large number of samples to consolidate, which is costly and should not be done after every task. Learning new tasks quickly in the STM and holding them till sleep provides a speed advantage and allows learning quickly with only periodic consolidation.

The dual memory architecture is a critical design choice for scalability and has also emerged naturally in humans, in the form of the complementary learning systems and the need to sleep periodically. Even though sleeping is a dangerous behavior for any organism since it can be harmed or attacked by a predator while asleep, sleep has still survived through eons of evolution and never been lost (Joiner, 2016). Today, most organisms with even a slightly developed nervous system (centralized or diffuse) display either sleep or light-resting behavior (Nath et al., 2017). The experiment demonstrates the importance of sleep, since without the dual memory architecture intertwined with periodic sleep, learning would be very short lived and highly time consuming (as in DGR).

## 5 ANALYSIS AND DISCUSSION

In this section we show that DGDMN shares some more remarkable characteristics with the human memory and present a discussion of some more related ideas. Due to space constraints, visualizations of the learnt latent structures when training jointly vs. sequentially have been deferred to appendix A. The hyperparameters of DGDMN ($\kappa$ and $n_{STM}$) have intuitive interpretations and we have provided simple heuristics to choose them without any complex searches (in appendix B).

**Resilience to noise and occlusion**: We use a VAE to be able to reconstruct representations of samples. Reconstructed images are less noisy and can recover from partial occlusion, which gives our model human-like abilities to recognize objects in noisy, distorted or occluded images. We test our LTM model and a NN model by jointly training on uncorrupted Digits data and testing on noisy and occluded images. We see that the LTM is more robust to noisy and occluded images and exhibits smoother degradation in classification accuracy because of its denoising reconstructive properties (see figure 8).

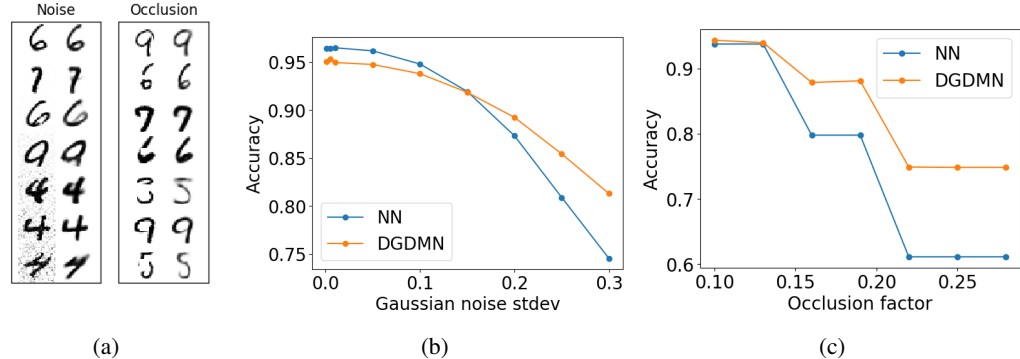

Figure 8: (a) LTM reconstruction from noisy and occluded digits, (b) Classification accuracy with increasing gaussian noise, and (c) Classification accuracy with increasing occlusion factor.

**The choice of underlying generative model**: Our consolidation ability and retention performance relies heavily on the generation and reconstruction ability of the underlying generative model. We chose a VAE for its reconstructive capabilities but our architecture is agnostic to the choice of the underlying generative model as long as the generator can generate reliable samples and reconstruct incoming samples accurately. Hence, variants of Generative Adversarial Networks (GAN) Goodfellow

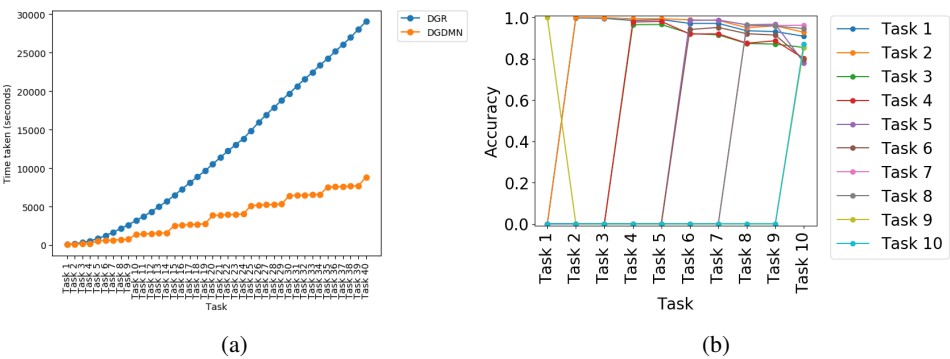

(a)                                             (b)

Figure 9: (a) Training time for DGDMN and DGR, (b) Accuracy curves: DGDMN (no STM).

et al. (2014) like BiGANs (Donahue et al., 2017), ALI (Dumoulin et al., 2017) and AVB (Mescheder et al., 2017) can also be used for the generative model depending on the modeled domain.

**Why use short-term memory?**: Our LTM always learns from STTMs and never from real data, and the STTMs' errors slowly propagate into the LTM and contribute to forgetting. An alternative could be to directly store data from new incoming tasks, consolidate it into the LTM after periodic intervals, and then discard the data. We show the accuracy curves on Digits dataset for this approach in figure 9b. This results in higher retention compared to DGDMN in figure 4 because LTM now learns from real data. However, this approach is not truly online since recently learnt tasks cannot be used immediately until after a sleep phase. Since the STM's error can be made smaller by using high capacity generators and classifiers, we suggest using a STM for true online continual learning.

**Connections to knowledge distillation**: Previous works on (joint) multitask learning have also proposed approaches to learn individual tasks with small networks and then "distilling" them jointly into a larger neural network (Rusu et al., 2015). Such distillation can sometimes improve performance on individual tasks if they share structure and at other times mitigate inter-task interference due to refinement of learnt functions while distilling (Parisotto et al., 2016). Though we do not use temperature-controlled soft-labels while consolidating tasks into the LTM (unlike distillation), we surmise that due to refinement and compression during consolidation phase, DGDMN is also able to learn joint task structure effectively while mitigating interference between tasks.

**Approaches based on synaptic consolidation**: Though our architecture draws inspiration from complementary learning systems and experience replay in the human brain, there is also considerable neuroscientific evidence for synaptic consolidation in the human brain (like in EWC). It might be interesting to explore how synaptic consolidation can be incorporated in our dual memory architecture without causing stagnation and we leave this to future work. We also plan to extend our architecture to learning optimal policies over time via reinforcement learning without explicit replay memories.

## 6 CONCLUSION

In this work, we have developed a model capable of learning continuously on sequentially incoming tasks, while averting catastrophic forgetting. Our model employs a dual memory architecture to emulate the complementary learning systems (hippocampus and the neocortex) in the human brain and maintains a consolidated long-term memory via generative replay of past experiences. We have shown that generative replay performs the best for long-term performance retention even for neural networks with small capacity, while demonstrating the benefits of using generative replay and a dual memory architecture via our experiments. Our model hyperparameters have simple interpretations and can be set without much tuning. Moreover, our architecture displays remarkable parallels with the human memory system and provides useful insights about the connection between sleep and learning in humans.

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

## 7 APPENDIX A

### 7.1 DEEP GENERATIVE REPLAY

---

**Algorithm 1:** Deep Generative Replay

---

1: **Input:** Current parameters of DGM, new samples: $(X, Y)$, dictionary for new samples: $D_{tasks}$ (there can be multiple tasks), minimum fraction: $\kappa$, memory capacity: $N_{max}$
2: **Output:** New parameters of DGM
    `// Compute sampling fractions`
3: $\eta_{tasks} := \frac{\sum D_{tasks}}{\sum D_{dgm} + \sum D_{tasks}}$ and $\eta_{gen} := 1 - \eta_{tasks}$
4: **if** $\eta_{tasks} < \kappa |D_{tasks}|$ **then**
5:    $\eta_{tasks} := \kappa |D_{tasks}|$ and $\eta_{gen} := 1 - \eta_{tasks}$
6: **end if**
    `// Compute number of samples`
7: **if** $|X| > \eta_{tasks} \times N_{max}$ **then**
8:    $n_{tasks} := \eta_{tasks} \times N_{max}$ and $n_{gen} := N_{max} - n_{tasks}$
9:    Subsample $(X, Y)$ to meet size $n_{tasks}$
10: **else**
11:    $n_{tasks} := |X|$ and $n_{gen} := \frac{\eta_{gen}}{\eta_{tasks}} \times |X|$
12: **end if**
    `// Generate and reconstruct samples`
13: Generate $n_{gen}$ samples: $X_{gen}$ from generator $G$ and labels from learner $L$: $Y_{gen} = L(X_{gen})$
14: $X_{recon}$ = Reconstruct $\{X, X_{gen}\}$ using the generator $G$
    `// Train the DGM`
15: Train the generator $G$ on $X_{recon}$
16: Train the learner $L$ on $(X_{recon}, \{Y, Y_{gen}\})$

---

Deep Generative Replay (algorithm 1), as described in section 3.1, consolidates new tasks for a DGM with previously learnt tasks. It first computes sampling fractions for new tasks ($\eta_{tasks}$) and previously learnt tasks ($\eta_{gen}$) and ensures a minimum fraction ($\kappa$) per new task (lines 3–6). Then it computes the number of samples to generate from previous tasks and whether to subsample the incoming task samples to satisfy the memory capacity $N_{max}$ (lines 7–12). Finally, it generates the required number of samples from previous tasks, reconstructs all data and trains the DGM on resulting data (lines 13–16). For a dictionary $D$, $\sum D$ is the total number of tasks in $D$ counting repetitions, while $|D|$ is the total number of tasks without repetitions. $|X|$ is the number of samples in set $X$.

Shin et al. (2017) have recently proposed a similar idea independently and Mocanu et al. (2016) have also employed a generative replay in two-layer restricted boltzmann machines, but they do not describe balancing new and generated samples and cannot recognize repeated tasks (section 4.2). Their generative replay without a dual memory architecture is costly to train (section 4.3) and a lack of reconstruction for new samples makes their representations less robust to noise and occlusions (section 5).

### 7.2 MORE EXPERIMENTS WITH ACCURACY AND FORGETTING CURVES

In this section, we present more experiments on the Shapes and the Hindi dataset, which contain sequences of tasks with geometric shapes and hindi consonants recognition respectively. We observed similar forgetting patterns as on the Digits dataset in section 4. All baselines exhibited catastrophic forgetting on these sequences of tasks, but DGR and DGDMN were able to learn the task structure sequentially (figures 10, 11). The same is reflected in the average forgetting curves in figure 12.

### 7.3 JOINTLY VS. SEQUENTIALLY LEARNT STRUCTURE

To explore whether learning tasks sequentially results in a similar structure as learning them jointly, we visualized t-SNE (Maaten & Hinton, 2008) embeddings of the latent vectors of the LTM generator (VAE) in DGDMN after training it: (a) jointly over all tasks (Figure 13a), and (b) sequentially over

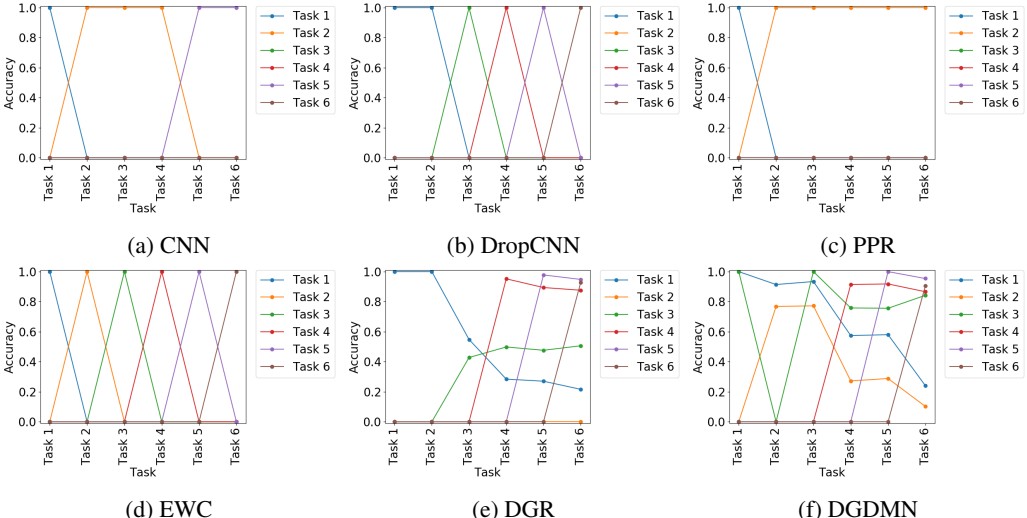

Figure 10: Accuracy curves for Shapes (x: tasks seen, y: classification accuracy on task).

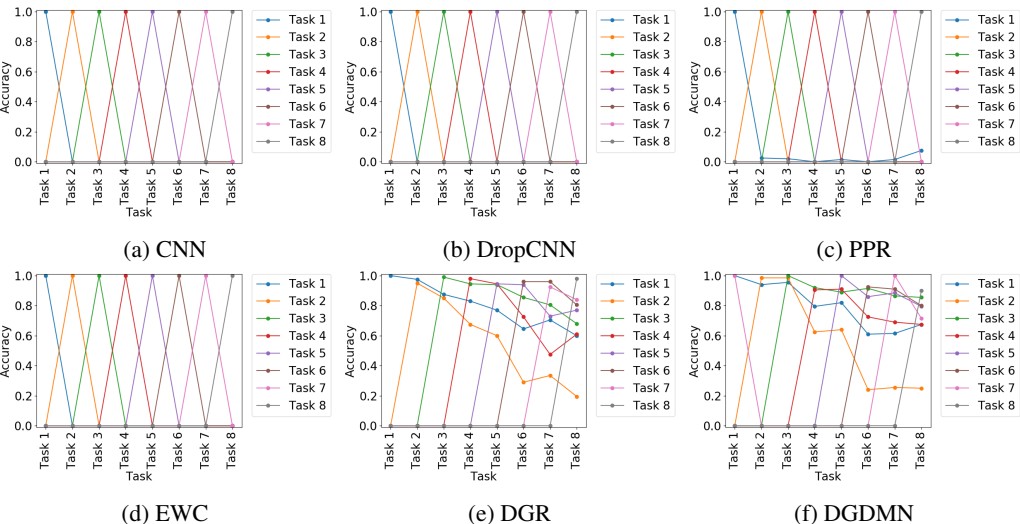

Figure 11: Accuracy curves for Hindi (x: tasks seen, y: classification accuracy on task).

tasks seen one at a time (Figure 13b) on the Digits dataset. To maintain consistency, we used the same random seed in t-SNE for both joint and sequential embeddings.

We observe that the LTM's latent space effectively segregates the 10 digits in both cases (joint and sequential). Though the absolute locations of the digit clusters differ in the two plots, the relative locations of digits share some similarity between both plots i.e. the neighboring digit clusters for each cluster are roughly similar. This may not be sufficient to conclude that the LTM discovers the same latent representation for the underlying shared structure of tasks in these cases and we leave a more thorough investigation to future work.

## 7.4 VISUALIZATIONS FOR THE JOINTLY AND SEQUENTIALLY LEARNT LTM

We also show visualizations of digits from the LTM when trained jointly on Digits tasks (Figure 14a) and when trained sequentially (Figure 14b). Though the digits generated from the jointly trained LTM are quite sharp, the same is not true for the sequentially trained LTM. We observe that the

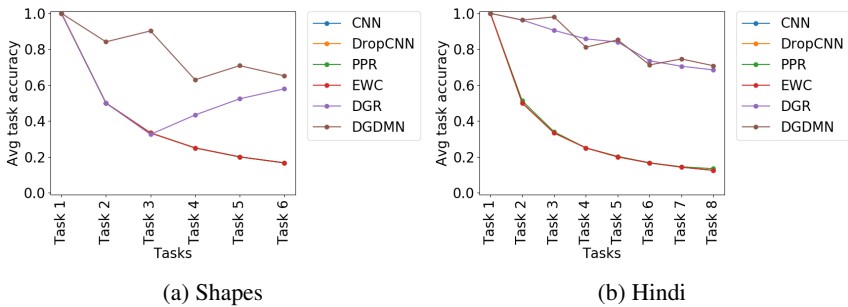

(a) Shapes                    (b) Hindi

Figure 12: Forgetting curves on Shapes and Hindi dataset (x: tasks seen, y: avg classification accuracy on tasks seen).

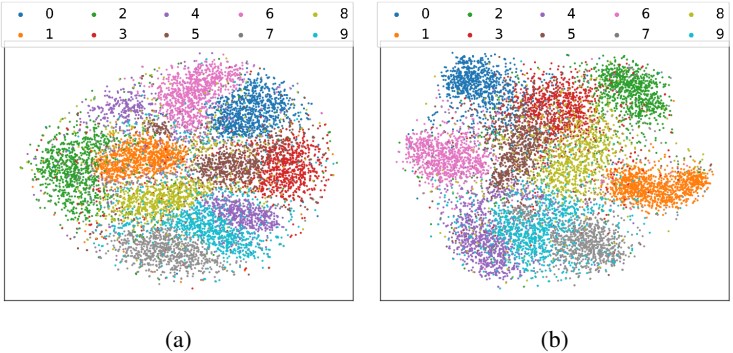

(a)                    (b)

Figure 13: t-SNE embedding for latent vectors of the VAE generator on Digits dataset when: (a) tasks are learnt jointly, and (b) tasks are learnt sequentially.

sequentially trained LTM produces sharp samples of the recently learnt tasks (digits $6, 7, 8$ and $9$), but blurred samples of previously learnt tasks, which is due to partial forgetting on these previous tasks.

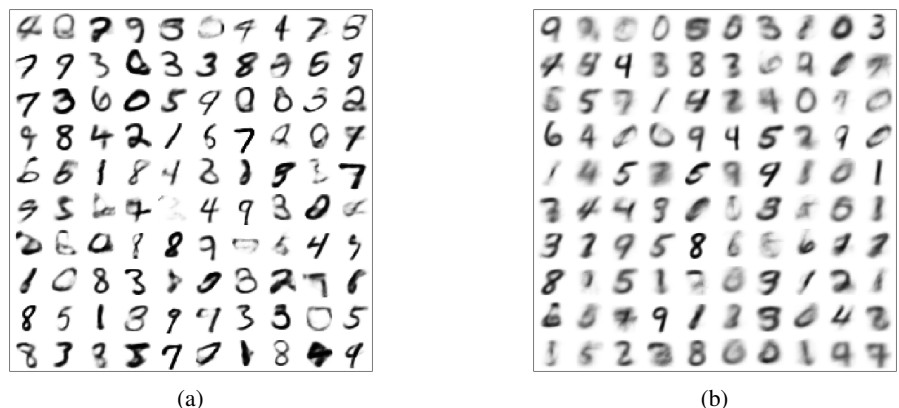

(a)                    (b)

Figure 14: Visualization of digits from LTM when trained: (a) jointly, (b) sequentially

# 8 APPENDIX B

## 8.1 DATASET PREPROCESSING

All our datasets have images with intensities normalized in the range $[0.0, 1.0]$ and size $(28 \times 28)$, except Hindi which has $(32 \times 32)$ size images.

**Permnist**: Our version involved six tasks, each containing a fixed permutation on images sampled from the original MNIST dataset. We sampled $30,000$ images from the training set and all the $10,000$ test set images for each task. The tasks were as follows: (i) Original MNIST, (ii) 8x8 central patch of each image blackened, (iii) 8x8 central patch of each image whitened, (iv) 8x8 central patch of each image permuted with a fixed random permutation, (v) 12x12 central patch of each image permuted with a fixed random permutation, and (vi) mirror images of MNIST. This way each task is as hard as MNIST and the tasks share some common underlying structure.

**Digits**: We introduce this smaller dataset which contains 10 tasks with the $t^{th}$ task being classification of digit $t$ from the MNIST dataset.

**TDigits**: We introduced a transformed variant of MNIST containing all ten digits, their mirror images, their upside down images, and their images when reflected about the main diagonal making a total of $40$ tasks. This dataset poses similar difficulty as the Digits dataset and we use it for experiments involving longer sequence of tasks.

**Shapes**: This dataset was extracted from the Quick, Draw! dataset recently released by Google (2017), which contains 50 million drawings across 345 categories of hand-drawn images. We subsampled $4,500$ training images and $500$ test images from all geometric shapes in Quick, Draw! (namely circle, hexagon, octagon, square, triangle and zigzag).

**Hindi**: Extracted from the Devanagri dataset (Kaggle, 2017) and contains a sequence of $8$ tasks, each involving image classification of a hindi language consonant.

## 8.2 Training algorithm and its parameters

All models were trained with RMSProp (Hinton, 2012) using learning rate $= 0.001$, $\rho = 0.9$, $\epsilon = 10^{-8}$ and no decay. We used a batch size of $128$ and all classifiers were provided 20 epochs of training when trained jointly, and 6 epochs when trained sequentially over tasks. For generative models (VAEs), we used gradient clipping in RMSProp with `clipnorm=` 1.0 and `clipvalue=` 0.5, and they were always trained for 25 epochs regardless of the task or dataset involved.

## 8.3 Neural network architectures

We chose all our models by first training them jointly on all tasks in a dataset to ensure that our models had enough capacity to perform reasonably well on all tasks. But we gave preference to simpler models over very high capacity models.

**Classifier Models**: Our implementation of NN, DropNN, PPR, EWC, learner for DGR and the learner for LTM in DGDMN used a neural network with three fully-connected layers with the number of units tuned differently according to the dataset ($24, 24$ units for Digits, $48, 48$ for Permnist and $36, 36$ for TDigits). DropNN also added two dropout layers, one after each hidden layer with droput rate = 0.2 each. The classifiers (learners) for Shapes and Hindi datasets had two convolutional layers ($12, 20 : 3 \times 3$ kernels for Shapes and $24, 32 : 3 \times 3$ kernels for Hindi) each followed by a $2 \times 2$ max-pooling layer. The last two layers were fully-connected ($16, 6$ for Shapes and $144, 36$ for Hindi). The hidden layers used ReLU activations, the last layer had a softmax activation, and the model was trained to minimize the cross-entropy objective function. The learners for STTMs in DGDMN were kept smaller for speed and efficiency concerns.

**Generative models**: The generators (VAE) for DGR and LTM of DGDMN employed encoders and decoders with two fully connected hidden layers each with ReLU activation for Permnist, Digits and TDigits, and convolutional variants for Shapes and Hindi. The sizes and number of units/kernels in the layers were tuned independently for each dataset with an approximate coarse grid-search. The size of the latent variable $z$ was set to 32 for Digits, 64 for Permnist, 96 for TDigits, 32 for Shapes and 48 for Hindi. The STTM generators for DGDMN were kept smaller for speed and efficiency.

## 8.4 Hyperparameters of DGDMN

DGDMN has two new hyperparameters: (i) $\kappa$: minimum fraction of $N_{max}$ reserved for incoming tasks, and (ii) $n_{STM}$: number of STTMs (also sleep/consolidation frequency). Both these have straightforward interpretations and can be set directly without complex hyperparameter searches.

$\kappa$ ensures continual incorporation of new tasks by guaranteeing them a minimum fraction of LTM samples during consolidation. Given that LTM should perform well on last $K$ tasks seen in long

task sequence of $T$ tasks, we observed that it is safe to assume that about $50\%$ of the LTM would be crowded by the earlier $T - K$ tasks. The remaining $0.5$ fraction should be distributed to the last $K$ tasks. So choosing $\kappa = \frac{0.5}{K}$ works well in practice (or as a good starting point for tuning). We made this choice in section 4.3 with $K = 10$ and $\kappa = 0.05$, and hence plotted the average accuracy over the last 10 tasks as a metric.

$n_{STM}$ controls the consolidation cycle frequency. Increasing $n_{STM}$ gives more STTMs, less frequent consolidations and hence a learning speed advantage. But this also means that fewer samples of previous tasks would participate in consolidation (due to maximum capacity $N_{max}$ of LTM), and hence more forgetting might occur. This parameter does not affect learning much till the LTM remains unsaturated (i.e. $N_{max}$ capacity is unfilled by generated + new samples) and becomes active after that. For long sequences of tasks, we found it best to keep at least $75\%$ of the total samples from previously learnt tasks to have appropriate retention. Hence, $n_{STM}$ can be set as approximately $\frac{0.25}{\kappa}$ in practice (as we did in section 4.3), or as a starting point for tuning.

## 8.5    ALGORITHM SPECIFIC HYPERPARAMETERS

**PPR**: We used a maximum memory capacity of about $3 - 6$ times the number of samples in a task for the dataset being learnt on (i.e. $18,000$ for Digits, $60,000$ for Permnist, $15,000$ for Shapes and $5,400$ for Hindi). While replaying, apart from the task samples, the remaining memory was filled with random samples and corresponding labels.

**EWC**: Most values of the coefficient of the Fisher Information Matrix based regularizer between $1$ to $500$ worked reasonably well for our datasets. We chose $100$ for our experiments.

**DGR and DGDMN**: $N_{max}$ for the DGM in DGR and for the LTM in DGDMN for Digits, Permnist, Shapes and Hindi was set as the total number of samples in the datasets (summed over all tasks) to ensure that there was enough capacity to regenerate the datasets well. For TDigits, we deliberately restricted memory capacity to see the effects of learning tasks over a long time and we kept $N_{max}$ as half the total number of samples. $n_{STM}$ was kept at 2 for Digits, Permnist and Shapes, 5 for TDigits and 2 for Hindi. $\kappa$ was set to be small, so that it does not come into play for Digits, Permnist, Shapes and Hindi since we already provided memories with full capacity for all samples. For TDigits, we used $\kappa = 0.05$ which would let us incorporate roughly 10 out of the 40 tasks well.

