# OpenReview forum: "Deep Generative Dual Memory Network for Continual Learning"
_ICLR.cc/2018/Conference — Reject_

### Official Review · AnonReviewer3 · 2017-11-27
**It's not clear if it is solving the core issue of catastrophic forgetting**

**Rating:** 5
**Confidence:** 4

**Review:**

This paper propose a variant of generative replay buffer/memory to overcome catastrophic forgetting. They use multiple copy of their model DGMN as short term memories and then consolidate their knowledge in a larger DGMN as a long term memory.

The main novelty of this work are 1-balancing mechanism for the replay memory. 2-Using multiple models for short and long term memory. The most interesting aspect of the paper is using a generate model as replay buffer which has been introduced before. As explained in more detail below, it is not clear if the novelties  introduced in this paper are important for the task or if they are they are tackling the core problem of catastrophic forgetting.

The paper claims using the task ID (either from Oracle or from a HMM) is an advantage of the model. It is not clear to me as why is the case, if anything it should be the opposite. Humans and animal are not given task ID and it's always clear distinction between task in real world.

Deep Generative Replay section and description of DGDMN are written poorly and is very incomprehensible. It would have been more comprehensive if it was explained in more shorter sentences accompanied with proper definition of terms and an algorithm or diagram for the replay mechanism.

Using the STTM during testing means essentially (number of STTM) + 1 models are used which is not same as preventing one network from catastrophic forgetting.

Baselines: why is Shin et al. (2017) not included as one of the baselines? As it is the closet method to this paper it is essential to be compared against.

I disagree with the argument in section 4.2.  A good robust model against catastrophic forgetting would be a model that still can achieve close to SOTA.  Overfitting to the latest task is the central problem in catastrophic forgetting which this paper avoids it by limiting the model capacity.

12 pages is very long, 8 pages was the suggested page limit. It’s understandable if the page limit is extend by one page, but 4 pages is over stretching.

---

> ### Author Response · Authors · 2017-12-27
> **Clarifications for AnonReviewer3**
>
> We are grateful for the valuable feedback. Below is our response for the questions and comments:
>
> 1- From the reviewer’s feedback, we felt that some of our contributions might have gone unnoticed and we clarify them here:
> Our architecture was inspired from the mammalian brain's dual memory architecture and the evidence for replay in the human brain. Though neuroscientific theories of complementary learning systems have existed for a long time, there is no clear agreement on why the brain has evolved a dual memory architecture and on the connection between sleep and learning? It is also unclear how the two memories interact (there is some evidence for some kind of experience replay).
> Our work made one of the first attempts at finding a plausible computational architecture solution. Apart from establishing why replay must be a generative process and showing the scalability and performance retention offered by a dual-memory architecture, our work also sheds light on the evolution of sleep and how it might be required for learning scalably. We do not claim that our architecture is exactly how the human brain functions, but our approach has remarkably similar characteristics to those observed for the human memory observed in neuroscience and psychology literature. Our work lays foundation for a neuroscience inspired solution to challenges in artificial general intelligence.
> At the same time, from an algorithmic perspective, our approach achieves many desirable characteristics absent in other state-of-the-arts: (a) no stagnation unlike in approaches which modulate neural plasticity or generate sparse representations, (b) permits gradual forgetting when lacking capacity to learn all tasks, (c) reconstruction and denoising capabilities, (d) works well under revision, and (e) works even for small neural networks and on datasets like Digits (with heavily correlated samples per task) where most other baselines undergo severe catastrophic forgetting.
>
> 2- We do not claim that using task IDs is an advantage of our model. It is a requirement, but is not particularly limiting since it can be handled in practice with a HMM-based inference scheme as has also been used by previous state-of-the-arts (e.g. Kirkpatrick et al., 2017).
>
> 3- We do not mitigate catastrophic forgetting in a **single** network, but rather in an architecture capable of learning continuously on incoming tasks.
>
> 4- As of writing this response, neither the arXiv version nor the NIPS version of Shin et al. (2017) contains any details of the network architectures or hyperparameters for any experiments (no supplementary material or github links either). Further, no details have been provided about the mixing ratio of samples to perform generative replay, and it is unclear how to re-implement their work. Even so, one of our baselines (DGR) is fairly close to their work and our approach outperforms DGR, both in terms of performance retention and training time (section 4.4).
>
> 5- Limiting model capacity does not get around the central challenge of catastrophic forgetting, but rather takes it head-on. See figure 3b of Kirkpatrick et al (2017) which shows that even after learning 10 tasks sequentially, their baseline (SGD+Dropout) drops to only little below 80% net accuracy on these tasks. Such forgetting can hardly be deemed catastrophic, and occurs because using large networks partly mitigates the problem. Using 2-hidden layer networks with above 400 units per layer (Goodfellow et al, 2015; Kirkpatrick et al, 2017) masks the contribution of any approach since the overparameterized network might be aiding in mitigating catastrophic forgetting. Our experiments show that our approach still retains a good accuracy even with small networks (two fully-connected layers having 48, 48 units; appendix B), whereas most other baselines are not able to retain their accuracy (see figures 4 and 5). If smaller models would have helped our approach, they would **also have helped the baselines**, which is clearly not the case (figure 4).
>
> 6- The reviewer might have misunderstood our comment about achieving less accuracy than SOTA. We only implied that we have not used large overparameterized networks for greater than 99% **joint** accuracies, but rather those with a reasonable (94-96%) joint accuracy (see point 5 for reason). We do indeed outperform the SOTA baselines in mitigating catastrophic forgetting, as shown by all our experiments. We have rectified this in the new draft to avoid misunderstanding for future readers.

---

> > ### Comment · AnonReviewer3 · 2018-01-12
> > **Responce**
> >
> > Using inspirations from brain architecture and mechanism is certainly great direction and I applause authors for good referencing of some of the relevant literature . But I get a sense from the authors both in the paper and in their comment that they are over emphasizing the link of their work and how brain functions. We should be more careful and cautious in drawing conclusions and connecting theories specially about the a topic like memory and sleep which is still heavily under investigation in neuroscience and cognitive science communities and no widely accepted there exist. The theory of short-term long-term memory and memory consolidate is not the only model for memory, e.g. there is the multiple trace model and there are more recent studies that question the memory consolidation models.
> >
> > I like to thank you the authors for adding new experiments and revising the paper. It has improved compared to the original version. I very much like the directions of this kind of research. But unfortunately I think it’s a complicated training mechanism with weak experimental evidence. Even though whole structure is novel, the components, and the breakdown of training to sleep, wake, or generative memory are novel.

---

> ### Author Response · Authors · 2017-12-27
> **Page limit and revised draft summary**
>
> 7- We had many insightful experiments in our paper and hence needed more than 8 pages. But respecting the reviewer’s advice, we have worked hard to shorten the paper and bring the main body closer to the recommended number of pages (at 9 pages). During the process we have also improved the figures, made some sections (especially section 3) more concise and understandable, and moved some parts to the appendices. We hope the reviewer will not object to our usage of an extra page considering the experiments and insights involved, and also since ICLR has no strict limit.
>
> 8- Below is a brief summary of changes in the new version of our paper:-
> 	[a] Figures 1 and 2 redrawn for clarity.
> 	[b] Length of main paper shortened from 12 to 9 pages, some parts moved to appendices.
> 	[c] Section 3 made more comprehensible (per your suggestion).
> 	[d] Deep Generative Replay added as an algorithm (appendix A), with clearer explanation.
> 	[e] Section 5 made more concise, discussion on MTL, task interference and distillation added.
> 	[f] Add results on two more datasets (Shapes, Hindi) to appendix A.
> 	[g] Some minor spelling and grammar issues rectified.
>
> Lastly, we thank the reviewer for taking the time to read our response and sincerely hope that in the light of the above clarifications, the reviewer would reconsider his/her rating.

---

### Official Review · AnonReviewer1 · 2017-11-28
**Successful sequential learning of MNIST task variants in challenging single pass setting**

**Rating:** 6
**Confidence:** 4

**Review:**

This paper reports on a system for sequential learning of several supervised classification tasks in a challenging online regime. Known task segmentation is assumed and task specific input generators are learned in parallel with label prediction. The method is tested on standard sequential MNIST variants as long as a class incremental variant. Superior performance to recent baselines (e.g. EWC) is reported in several cases. Interesting parallels with human cortical and hippocampal learning and memory are discussed.

Unfortunately, the paper does not go beyond the relatively simplistic setup of sequential MNIST, in contrast to some of the methods used as baselines. The proposed architecture implicitly reduces the continual learning problem to a classical multitask learning (MTL) setting for the LTM, where (in the best case scenario) i.i.d. data from all encountered tasks is available during training. This setting is not ideal, though. There are several example of successful multitask learning, but it does not follow that a random grouping of several tasks immediately leads to successful MTL. Indeed, there is good reason to doubt this in both supervised and reinforcement learning domains. In the latter case it is well known that MTL with arbitrary sets of task does not guarantee superior, or even comparable performance to plain single-task learning, due to ‘negative interference’ between tasks [1, 2]. I agree that problems can be constructed where these assumptions hold, but this core assumption is limiting. The requirement of task labels also rules out important use cases such as following a non-stationary objective function, which is important in several realistic domains, including deep RL.


[1] Parisotto, Emilio; Lei Ba, Jimmy; Salakhutdinov, Ruslan:
Actor-Mimic: Deep Multitask and Transfer Reinforcement Learning. ICLR 2016.
[2] Andrei A. Rusu, Sergio Gomez Colmenarejo, Çaglar Gülçehre, Guillaume Desjardins, James Kirkpatrick, Razvan Pascanu, Volodymyr Mnih, Koray Kavukcuoglu, Raia Hadsell: Policy Distillation. ICLR 2016.

---

> ### Author Response · Authors · 2017-12-27
> **Clarifications for AnonReviewer1**
>
> We are grateful for the valuable feedback. Below is our response for the questions and comments:
>
> 1- Since this was an initial attempt to understand the memory architecture in mammals and design a similar one to mitigate forgetting and learn continuously, we experimented with simple tasks in the supervised settings (and unsupervised settings, since our VAEs do unsupervised reconstruction). Nevertheless, our experiments were quite insightful (at least for us) and have provided interesting ways to explore this approach further. Since you mentioned experimenting only with sequential MNIST variants, we have also added experiments with two new datasets to the current draft (one dealing with learning geometric shapes and other for hindi language) to clarify that our approach has no reliance on MNIST variants and easily extends beyond it.
>
> The revered reviewer also pointed out that our algorithm requires task IDs, which may not be available. We emphasize that in supervised and unsupervised settings, where tasks come in batches, this is not particularly limiting since IDs can be generated via a task identification module in practice (say using a HMM-based inference scheme), as has been done by previous work (e.g. EWC: Kirkpatrick et al., 2017).
>
> However, we eventually wish to scale up the architecture to continually streaming inputs like in reinforcement learning. Even in this setting if the idea is to learn on multiple RL domains sequentially, then our method can be extended easily with task IDs (as done by Kirkpatrick et al., 2017). However, learning without forgetting within a single domain is a somewhat more challenging job and in such setting, a replacement for task IDs might be required. We point out that we this was out of scope of our current work and should not be counted as a shortcoming, but we are actively working towards the same for future work.
>
> 2- We understand that MTL and inter-task interaction are important subproblems in the field of continual learning, but we focus on mitigating catastrophic forgetting in this work. This is also an equally important problem and it is hard to fit two broad problems in a single paper. Modeling inter-task interaction deserves a study of its own. As mentioned at the end of section 2, our goal is to learn tasks sequentially, while avoiding catastrophic forgetting and achieve test loss close to that of a jointly trained model.
> As for distillation, authors in [2] write: "Policy distillation may offer a means of combining multiple policies into a single network without the damaging interference and scaling problems. Since policies are compressed and refined during the distillation process, we surmise that they may also be more effectively combined into a single network." This is also true of our approach. Since our generative replay distills several tasks together from the STM to the LTM while refining them along with previously existing tasks in the LTM, we believe that it provides a similar effective way to combine tasks and deal with the damaging inter-task interference.
> We have also added connections between our approach and distillation to the new version of our paper (section 5).
>
> 3- Below is a brief summary of changes in the new version of our paper:-
> 	[a] Figures 1 and 2 redrawn for clarity.
> 	[b] Length of main paper shortened from 12 to 9 pages, some parts moved to appendices.
> 	[c] Section 3 made more comprehensible (as suggested by AnonReviewer3).
> 	[d] Deep Generative Replay added as an algorithm (appendix A), with clearer explanation.
> 	[e] Section 5 made more concise, discussion on MTL, task interference and distillation added.
> 	[f] Add results on two more datasets (Shapes, Hindi) to appendix A.
> 	[g] Some minor spelling and grammar issues rectified.
>
> Lastly, we thank the reviewer for taking the time to read our response and hope that your queries were appropriately clarified.
>
> [2] Andrei A. Rusu, Sergio Gomez Colmenarejo, Çaglar Gülçehre, Guillaume Desjardins, James Kirkpatrick, Razvan Pascanu, Volodymyr Mnih, Koray Kavukcuoglu, Raia Hadsell: Policy Distillation. ICLR 2016.

---

### Official Review · AnonReviewer2 · 2017-11-30
**A moderately surprising but useful study on replay based learning**

**Rating:** 7
**Confidence:** 2

**Review:**

This paper introduces a neural network architecture for continual learning. The model is inspired by current knowledge about long term memory consolidation mechanisms in humans. As a consequence, it uses:
-	One temporary memory storage (inspired by hippocampus) and a long term memory
-	A notion of memory replay, implemented by generative models (VAE), in order to simultaneously train the network on different tasks and avoid catastrophic forgetting of previously learnt tasks.
Overall, although the result are not very surprising, the approach is well justified and extensively tested. It provides some insights on the challenges and benefits of replay based memory consolidation.

Comments:

1-	The results are somewhat unsurprising: as we are able to learn generative models of each tasks, we can use them to train on all tasks at the same time, a beat algorithms that do not use this replay approach.
2-	It is unclear whether the approach provides a benefit for a particular application: as the task information has to be available, training separate task-specific architectures or using classical multitask learning approaches would not suffer from catastrophic forgetting and perform better (I assume).
3-	So the main benefit of the approach seems to point towards the direction of what possibly happens in real brains. It is interesting to see how authors address practical issues of training based on replay and it show two differences with real brains: 1/ what we know about episodic memory consolidation (the system modeled in this paper) is closer to unsupervised learning, as a consequence information such as task ID and dictionary for balancing samples would not be available, 2/ the cortex (long term memory) already learns during wakefulness, while in the proposed algorithm this procedure is restricted to replay-based learning during sleep.
4-	Due to these differences, I my view, this work avoids addressing directly the most critical and difficult issues of catastrophic forgetting, which relates more to finding optimal plasticity rules for the network in an unsupervised setting
5-	The writing could have been more concise and the authors could make an effort to stay closer to the recommended number of pages.

---

> ### Author Response · Authors · 2017-12-27
> **Clarifications for AnonReviewer2**
>
> We are grateful for the valuable feedback. Below is our response for the questions and comments:
>
> 2- Classic MTL approaches do not involve task segmentation because they assume that all tasks are predefined (in which case assigning an ID is trivial and can be done manually without fear of repetition). Continuous learning requires task segmentation since tasks arrive sequentially, and we do not explicitly store any task samples. Task segmentation is not a very limiting assumption and can be met in practice using an HMM-based inference scheme as has been used in existing state-of-the-arts (e.g. EWC: Kirkpatrick et al., 2017). We have also added a discussion of connections to classical MTL and policy distillation in section 5.
>
> 3- In addition to shedding light into the utility of the brain's dual memory architecture, our method also achieves many other desirable characteristics absent from other state-of-the-arts (please see our joint note to reviewers).
> As for learning during wakefulness, we omitted it due to space constraints, since it is easily emulated by small intermediate consolidation steps with fewer training iterations.
>
> 4- Unsupervised plasticity is one way to mitigate catastrophic forgetting, but it is not the only way. Performing (unsupervised) generative replay along with dual-memory consolidation is a viable option and our experiments show that generative replay outperforms plasticity and sparse-representation oriented approaches (Kirkpatrick et al., 2017; Goodfellow et al., 2015).
>
> 5- We had many insightful experiments in our paper and hence needed more than 8 pages. But respecting the reviewer’s advice, we have worked hard to shorten the paper and bring the main body closer to the recommended number of pages (at 9 pages). During the process we have also improved the figures, made some sections more concise, and moved some parts to the appendices. We hope the reviewer will understand and not object to our usage of an extra page considering the experimentation and insights involved, and also since ICLR has no strict limit.
>
> 6- Below is a brief summary of changes in the new version of our paper:-
> 	[a] Figures 1 and 2 redrawn for clarity.
> 	[b] Length of main paper shortened from 12 to 9 pages, some parts moved to appendices.
> 	[c] Section 3 made more comprehensible (as suggested by AnonReviewer3).
> 	[d] Deep Generative Replay added as an algorithm (appendix A), with clearer explanation.
> 	[e] Section 5 made more concise, discussion on MTL, task interference and distillation added.
> 	[f] Add results on two more datasets (Shapes, Hindi) to appendix A.
> 	[g] Some minor spelling and grammar issues rectified.
>
> Lastly, we thank the reviewer for taking the time to read the response and hope that your queries were appropriately clarified.

---

### Public Comment · ~Siddhant_Jayakumar2 · 2017-11-28
**Clarification on tasks**

Interesting approach!

Just a few questions about the tasks and parameters.

The permuted MNIST variant considered here seems to be different from the setting in the Goodfellow et al (2014) and Kirkpatrick et al (2017) unless I'm mistaken? What was the rationale behind this? Does the model proposed also cope well with the standard "fixed random permutation of all pixels" for each task, as opposed to the cropping and whitening style tasks employed in the paper?

Further, how often was the "sleep" or consolidation phase used?

---

> ### Author Response · Authors · 2017-11-29
> **Yes, indeed! Our model works for fixed random permutations of all pixels too.**
>
> Our permuted MNIST variant does contain the "fixed random permutation of all pixels" tasks (see sec 8.1, appendix B, tasks iv and v for permnist). We basically included a few of all kinds of tasks (whitening-type, permutation-type and reflection-type), as was necessary to prove that our algorithm performs as well as the baselines on a set of tasks used in the past.
> However, permnist is not our major focus, since as pointed out in section 4.2.1, permnist is not a good dataset to test for catastrophic forgetting. We observed that it is easily conquered by most approaches if they use a largely overparameterized neural network. See figure 3b of Kirkpatrick et al (2017) and you'll observe that even after learning 10 tasks sequentially, their baseline (SGD+Dropout) drops to only little below 80% accuracy. Such forgetting can hardly be deemed catastrophic, and is partly because of using really large networks. Using a 2-hidden layer network each with above 400 units per layer (Goodfellow et al, 2015; Kirkpatrick et al, 2017) allows the network to essentially finds ways to **memorize** samples and labels from different tasks without inducing much parameter sharing. In such cases, it is unclear if it is the continual learning algorithm at work, or just the overparameterized network aiding in mitigating catastrophic forgetting. But our experiments show that our approach still retains a good accuracy even with small networks (with fully-connected layers having 48, 48 units; see appendix B), whereas most other baselines are not able to retain their accuracy (see figure 5a).
> Moreover, we show that datasets like Digits, although simple at first glance, are actually much more challenging datasets to test for catastrophic forgetting and are hard to conquer even with overparameterized networks. Hence, we focused most of our attention on Digits and TDigits.
> Lastly, to clarify again, we did experiment with the full permnist dataset and can include more "permutation" tasks if needed, since our algorithm (DGDMN) works perfectly well with all permutation-type tasks and outperforms all baselines on the full permnist too.
>
> The consolidation phase frequency is characterized by n_{STM} hyperparameter. n_{STM} was 2 for both Digits and Permnist, and 5 for TDigits (see section 8.4 in appendix B).

---

### Decision · Program_Chairs · 2018-01-29
**ICLR 2018 Conference Acceptance Decision**

**Decision:**

Reject

**Comment:**

Thank you for submitting you paper to ICLR. The big-picture idea is fairly simple, although the implementation is certainly challenging requiring a deep generative model to be trained as part of the final system. The experimental validation is not sufficient to warrant publication. A comparison to a larger number of competitors e.g. [1,2] on a greater range of tasks is required.

[1] Continual Learning Through Synaptic Intelligence Friedemann Zenke BenPoole SuryaGanguli, ICML 2017
[2] Gradient Episodic Memory for Continual Learning, David Lopez-Paz and Marc’Aurelio Ranzato, NIPS 2017